# Exploring neural entrainment and synchrony in response to repeated 60 Hz flickering white light in healthy volunteers

MohammadAmin Alamalhoda[1], Friederike Leesch[1], Francesca Giovanetti[1], Eoghan Dunne[2,3], Giuseppina Pilloni[4], Mark Caffrey[1,3], Jack O'Keeffe[1], Alessandro Venturino[1,5☉], Maria Teresa Ferretti[1☉*]

**1** Syntropic Medical GmbH, Klosterneuburg, Austria, **2** School of Medicine, University of Galway, Galway, Ireland, **3** Electrical and Electronic Engineering, School of Engineering, University of Galway, Galway, Ireland, **4** Department of Neurology, New York University Grossman School of Medicine, New York, New York, United States of America, **5** Institute of Science and Technology Austria (ISTA), Klosterneuburg, Austria

☉ These authors contributed equally to this work.
* mariateresa@syntropicmedical.com

## Abstract

Flickering light is a new promising, fully non-invasive brain stimulation technique that utilizes intermittent sensory stimulation to induce brainwave synchronization (entrainment). While the effects of 40 Hz externally induced neural entrainment have been extensively described, little is known about 60 Hz entrainment in humans. This study presents preliminary observations on the neural and somatic response to flickering 60 Hz light in healthy volunteers over a 3-week period. Fourteen volunteers were randomized to receive either 60 Hz flickering white light or constant light as sham (30-min sessions, 3 weeks, 5 days/week on weekdays). Neural entrainment was assessed with EEG on days 1, 5 and 19. Salivary cortisol and C-reactive protein (CRP) levels, measured with ELISA, assessed the somatic response to stimulation. Side effects and well-being were monitored via questionnaires. EEG recordings showed neural entrainment and synchrony in response to 60 Hz flickering light across multiple cortical regions, including occipital, central, temporal, and frontal areas. The entrainment power and synchronization between different cortical regions declined significantly by day 19 compared to day 1, indicating possible neural habituation. Cortisol and CRP salivary levels were unchanged, and minor side effects were reported with equal frequency in the active and sham groups. Our findings show that 60 Hz flickering light can induce significant neural entrainment and synchrony in healthy adults and is well tolerated. The decline in entrainment strength and neural synchrony observed with repeated 60 Hz stimulations suggests plastic changes in the cortex. To the best of our knowledge, this is the first study to characterize neural and somatic responses to repeated 60 Hz flickering visual stimuli. Given the well-known connection between 60 Hz brain oscillations and cognition, neuroplasticity,

**Data availability statement:** The EEG data and custom code used for preprocessing data and analysis are available at https://github.com/AminAlam/HVS, DOI: https://doi.org/10.5281/zenodo.17019688.

**Funding:** This study was funded by Syntropic Medical and supported by an Austria Wirtschaftsservice (AWS) grant (grant number P2414247 to Syntropic Medical). Syntropic Medical employees were involved in study design, data collection and analysis, decision to publish, and preparation of the manuscript. AWS had no role in study design, data collection and analysis, decision to publish, or preparation of the manuscript.

**Competing interests:** M.T.F., M.A., F.L., F.G. are employees of Syntropic Medical GmbH. A.V. discloses an international patent application (PCT/EP2020/079365). A.V., J.O. and M.C. are co-founders of Syntropic Medical GmbH. In the past 3 years, M.T.F. has received consultancy and speaking fees from Angelini Pharma, Eli Lilly, EPH health, and Biogen, unrelated to the present work. G.P. received consultancy and speaking fees from Ceragem and Soterix Medical Inc., with no relation to the present work. The other authors declare no competing interests.

and their role in neuropsychiatric disorders, additional research in both preclinical and clinical settings is warranted.

---

## 1. Introduction

*Neuromodulation* is a collection of techniques aimed at modulating diffuse neuronal activity to achieve therapeutic effects. When modulation is obtained through the application of external energy (e.g., electrical currents, magnetic fields, light, or ultrasound) to the brain, it is referred to as *neurostimulation* [1]. Non-invasive brain stimulation (NIBS) techniques, such as electroconvulsive therapy (ECT), transcranial magnetic stimulation (TMS), and transcranial electrical stimulation (tES), exert measurable structural and functional effects on the brain. These effects include increased neuroplasticity [2], changes in brain structure [3] and connectivity [4], and restoration of brain-derived neurotrophic factor levels [5]. Some NIBS protocols are FDA-approved for the treatment of various brain diseases, including depression [6] and obsessive-compulsive disorder [7], underscoring the therapeutic potential of neuromodulation.

*Emerging non-invasive neuromodulatory approaches* leverage different energy modalities to modify neuronal excitability and modulate brain activity. These include transcranial focused ultrasound [8], temporal interference stimulation [9], and thermal stimulation using near-infrared lasers [10].

*A promising new, fully non-invasive brain stimulation technique, utilizes intermittent sensory stimulation*. Brainwaves naturally synchronize with rhythmic external inputs – such as flickering lights, speech, music, or touch – a process known as neural entrainment [11]. Entrainment in the gamma frequency band (30–70 Hz) is especially interesting because gamma oscillations are linked to attention, cognition, and executive control [12], and alterations have been described in several neuropsychiatric conditions [13,14].

*Multisensory external stimulation* at 40 Hz, combining visual and acoustic stimuli, has been extensively studied in both mouse models [15] and humans [16]. 40 Hz stimulation has been shown to induce strong and well-tolerated neural entrainment. In models of Alzheimer's disease-like amyloidosis, 40 Hz stimulation reduced amyloid pathology [15,16], modulated microglial activity [15] and promoted glymphatic clearance of amyloid [17]. In humans, it increased connectivity between the precuneus and posterior cingulate cortex, suggesting neuroplastic effects [18].

Although 40 Hz stimulation has been extensively studied in the context of neural entrainment, particularly with regard to its therapeutic potential for Alzheimer's disease, the effects of 60 Hz stimulation in humans remain largely unexplored. However, accumulating evidence suggests that endogenous 60 Hz gamma oscillations are functionally linked to attention and cognitive control and may be altered in neuropsychiatric conditions, such as schizophrenia and depression [13,14]. Our previous preclinical work in mice demonstrated that visual stimulation with 60 Hz flickering light induces neural entrainment and juvenile-like neuroplasticity via microglia-mediated remodeling of perineuronal nets (PNNs) [19].

*A small number of studies have investigated the effects of non-invasive 60 Hz stimulation on cognition and neural synchrony in humans.* For instance, auditory stimulation at 60 Hz has been shown to reduce intrusion errors and improve cognitive control in healthy adults [20] while transcranial alternating current stimulation (tACS) at 60 Hz over the dorsolateral prefrontal cortex (DLPFC) has been associated with enhanced declarative long-term memory (LTM) [21].

These studies have primarily employed auditory or electrical stimulation. The use of visual 60 Hz flicker to induce neural entrainment, particularly with repeated sessions over multiple weeks, has not yet been systematically explored.

*Our study addresses this gap by evaluating the neural, somatic, and tolerability outcomes of repeated 60 Hz flickering light stimulation in healthy adults.* We applied LED-generated, 60 Hz pulsed flickering white light with a square wave modulation to a group of 14 healthy volunteers. Brain activity was measured via electroencephalography (EEG) on days 1, 5, and 19. Cortisol and C-reactive protein (CRP) levels were collected from saliva samples, and tolerability was assessed via participant questionnaires. Our preliminary results suggest that 60 Hz flickering light stimulation can induce strong and widespread neural entrainment and synchronization across multiple cortical regions. The stimulation was well tolerated by healthy participants, with no severe adverse events reported.

## 2. Methods

### 2.1 Ethical approval and overall study design

*Ethical Approval.* The study was conducted in accordance with the World Medical Association Declaration of Helsinki for human experimentation and applicable national laws. Ethical approval was obtained from the Lower Austria Ethical Commission (study approval number: GS3-EK-4/908–2024 approved on July 1, 2024). All procedures were carried out at Plöcking 1, XISTA Science Park, 3400 Klosterneuburg, Austria, under medical supervision. Written informed consent was obtained by study personnel from all participants.

*Blinding and group assignment.* Following eligibility screening, participants were randomly assigned (1:1 allocation ratio) to either the active stimulation group (60 Hz white light) or the sham group (constant light; **Fig 1B**). The study followed a single-blind design, where participants were unaware of their group assignment. To assess blinding efficacy, participants were asked at the end of the study to guess their group assignment.

*Stimulation schedule and protocol.* Each participant received one stimulation session per day, Monday through Friday for 3 consecutive weeks, totaling 15 sessions (see protocol in **Fig 1C**). All sessions were conducted during daylight hours (between 9:00 am and 4:00 pm); the time of day was kept consistent for each participant across all sessions.

On EEG recording days (day 1, day 5, and day 19), the session followed a structured, three-step 30 min protocol (**Fig 1B**):

| Step I | Baseline (No light) | 5 minutes with no light stimulation |
|---|---|---|
| Step II. | Constant light | 5 minutes of constant light exposure |
| Step III. | Stimulus light | Active group: 20 min of 60 Hz flickering light |
| | | Sham group: 20 min of constant light |

Saliva samples for biomarker analysis were collected on these days (1, 5, and 19). On non-EEG days, participants received a continuous 30-minute stimulation session, with 60 Hz flickering light exposure for the active group, or constant light for the sham group.

*Stimulation environment and setup.* Participants were seated comfortably on a chair with a headrest to provide additional support and ensure stability. The wearable device was carefully fitted on the participant's face to ensure both comfort and consistent placement. The environment was kept quiet, with dim ambient lighting (S1A Fig). The operator activated the controller and adjusted the light intensity (for each session) based on real-time verbal feedback from the

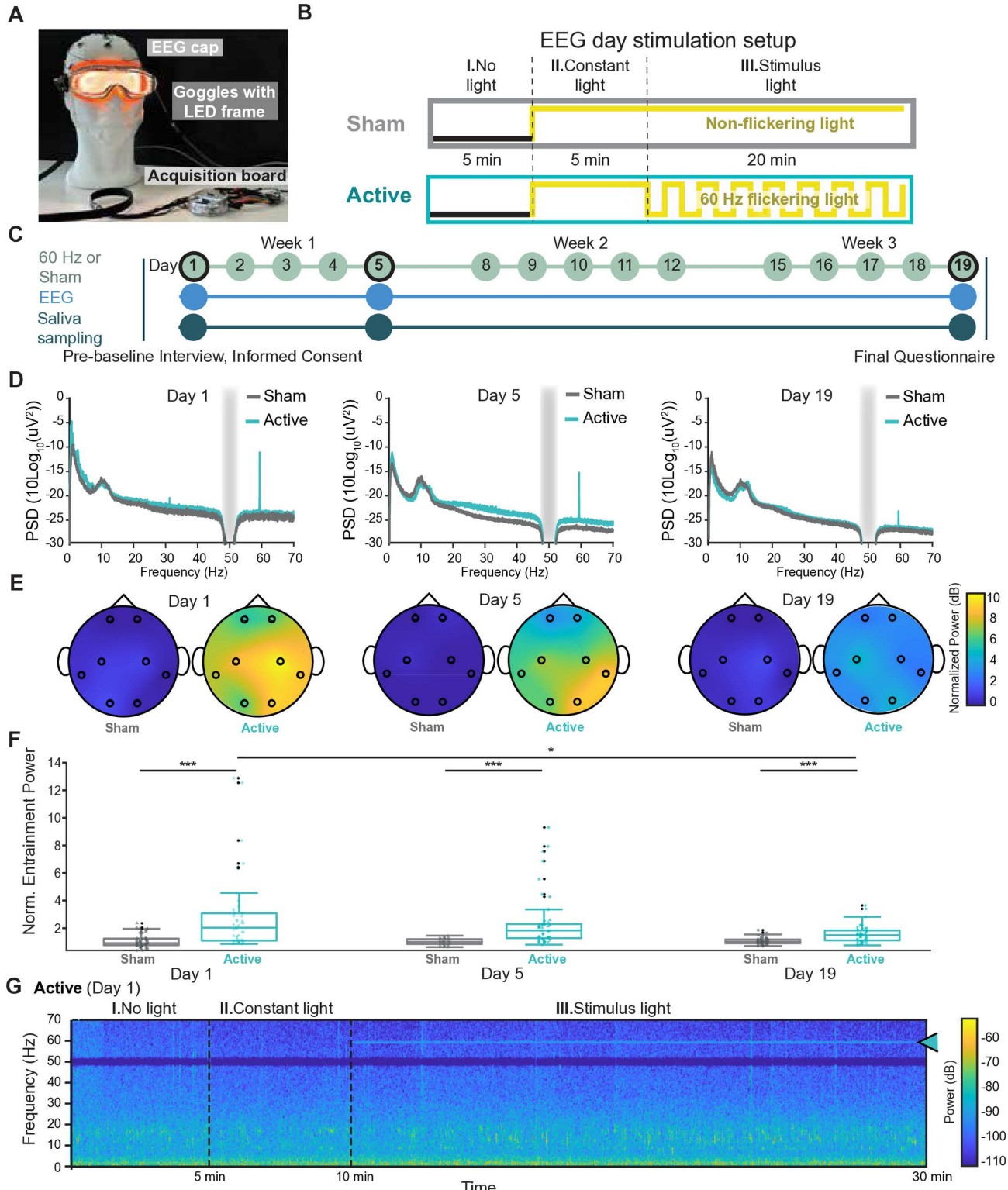

**Fig 1. 60 Hz-induced neural entrainment. (A)** Schematic of the experimental setup, showing a wearable device (adapted from a Uvex mask equipped with LED lights) designed to be compatible with an EEG setup. **(B)** Overview of each EEG session, which included two control phases (I. no light and II. constant light) and one stimulus phase (III. stimulus light **(C)** Diagram of the experimental timeline: participants underwent light stimulation over

three weeks, with EEG recordings and saliva sampling performed on days 1, 5, and 19 (indicated by black circles). On EEG days, the stimulation setup depicted in Panel B was applied. On non-EEG days (no black circles), participants received either an active stimulus (60 Hz flickering light) or a sham stimulus (constant light) for 30 minutes per session. **(D)** Scalp EEG power spectral density (PSD) averaged across all channels of all participants in each group. The gray bar indicates the 50 Hz line noise, which was notch-filtered. A strong 60 Hz frequency component is evident from start to finish but decreases over the course of the study at day 5 and day 19. **(E)** Topographic maps showing normalized changes in 60 Hz PSD (relative to baseline) averaged across participants of each group. **(F)** Significant differences in normalized changes in 60 Hz PSD values were observed across all channels between the active and sham groups on days 1, 5, and 19. Statistical significance for these inter-group comparisons was assessed using the Wilcoxon rank-sum test. Furthermore, within the active group, significant differences in normalized PSD were detected between day 1 vs. day 5 and day 1 vs. day 19, accounting for repeated measurements. These intra-group comparisons across days were evaluated using the Kruskal-Wallis test, followed by post-hoc pairwise comparisons performed using Dunn's test with Bonferroni correction. The normality of the data was assessed using the Shapiro-Wilk test, and non-parametric methods were applied due to deviations from normality. In this figure, different channels are shape-coded. $*=p<0.05$, $***=p<0.001$. Detailed p-values are provided in S3 Table. **(G)** Averaged Short-Time Fourier Transform (STFT) across all EEG channels of a representative active group participant, demonstrating visible 60 Hz entrainment during light stimulation, as indicated by the arrow.

participant to maximize comfort. Once started, all stimulation sessions (active and sham) were completed and there were no cases of volunteers asking to interrupt beforehand.

*Monitoring and data protection.* Daily side effects were assessed via self-report, and a final questionnaire was administered at the end of the study to investigate tolerability and blinding (see below). All data were pseudonymized and handled in accordance with national and international data protection regulations ensuring participant privacy throughout the study.

## 2.2 Participants

The study involved a cohort of 14 healthy young adults (see demographics in **Table 1**), recruited between July 23 and September 10, 2024. The main exclusion criteria were diagnosis of neurodegenerative or psychiatric diseases, in particular, history of seizures or epilepsy, migraine, and tinnitus. Full inclusion/exclusion criteria are provided in the S1 Table.

Participants were randomly assigned in a 1:1 ratio to either the active stimulation group (n = 8) or the sham stimulation (control) group (n = 6). The groups were balanced for self-reported sex, age, race, and education (**Table 1**).

## 2.3 Light stimulation device design

Manufacturer information and catalog numbers for components used to build the light stimulation prototype are listed in S2 Table.

*Headset.* To enable controllable and standardized flickering white light stimulation, we developed a custom wearable headset based on Uvex Ultrasonic safety glasses. The headset contained a 35 cm segment of a PowerLED Chromatic

**Table 1. Demographics of study participants.**

|  | Baseline | | | EEG sample | |
|---|---|---|---|---|---|
|  | Sham (n = 6) | Active (n = 8) | p-value | Active (n = 6) | p-value |
| **Age (n of years mean, SD)** | 25.5, 2.59 | 28.63, 6.37 | 0.6 | 29.83, 7.03 | 0.47 |
| **Female sex (n, %)** | 3, 50 | 5, 62.5 | 1 | 4, 67 | 1 |
| **Years of education (mean, SD)** | 18.83, 3.19 | 19.63, 3.78 | 0.84 | 20.67, 3.50 | 0.52 |
| **Race (n, %)** |  |  |  |  |  |
| Caucasian | 4, 66.67 | 8, 100 | 0.16 | 6, 100 | 0.45 |
| Asian | 2, 33.33 | 0, 0 |  | 0, 0 |  |

The table summarizes the main characteristics of the study participants at baseline and after two dropouts in the active group (sample used for EEG analysis). Sex and race were self-reported. Data were analyzed using a Wilcoxon Rank-sum test for age and years of education and Fisher's exact test for proportions. No statistically significant differences were observed across groups, also accounting for the two dropouts (p-values refer to comparison with sham group).

LED strip, arranged around the ocular perimeter to direct tangential light towards the periphery of the participants visual field (see **Fig 1A**).

*Light Control Circuit Design.* The 60 Hz band square signal was generated via an Arduino Nano board PWM pin soldered onto a Printed Circuit Board. The signal passed through a circuit comprising a 10 kΩ potentiometer for intensity adjustment and a p-channel MOSFET to regulate current delivery to the LEDs. To ensure fast and clean switching, the gate of the MOSFET was driven by a MOSFET gate driver.

*Power supply.* Power was supplied via an external power bank, positioned more than 1 m away from the subject to reduce electromagnetic interference.

*Synchronization Circuit Design.* To ensure the synchronization between the onset of the light stimulation and the EEG signal recording, a photoresistor was integrated to detect the light emitted from the LEDs. The analog signal from the photoresistor was converted into a digital signal using a second Arduino Nano board and transmitted to the acquisition board of the EEG system.

*Frequency calibration.* To avoid rounding errors inherent in generating an exact 60 Hz signal (16.6̲ ms), the stimulation frequency was set to 59.6 Hz (16.778 ms), allowing a precise and stable output. The LEDs emitted a square-wave function of white light at 60 Hz with a duty cycle of 50% (rise time: 1.5 ms, fall time: 3.5 ms). The stimulation frequency was manually verified using an oscilloscope before each data collection to ensure consistent frequency throughout the recording.

*Spectral characteristics.* The emitted light ranged from 440–770 nm, corresponding to the full spectrum of visible light colours, measured with a spectrometer (S1B Fig), with a color temperature of 4000 K. Intensity was adjustable between 74–94 µW (measured at 2 cm distance, ThorLabs PM100D), this range of intensities is considered safe to the eyes. The exact intensity was selected individually per participant to ensure a comfortable experience during each session based on the participant's verbal feedback. On average, subjects chose a light intensity of 80 µW (SD 7 µW across subjects), with low intra-individual variability (max SD 9 µW for any individual subject) (S1C, D Fig).

*Faraday cage.* To minimize electromagnetic interference during EEG recordings, the LED strip and its cable were enclosed within a copper mesh Faraday cage, all connected to the EEG system ground. The Faraday cage reduced electromagnetic emissions from 203 µT to 66 µT when measured at a distance of 10 cm from the LED strip using an MULTIFIELD-EMF450 meter.

## 2.4 EEG recording and data capture

EEG data were recorded using the OpenBCI Cyton + Daisy Biosensing Boards equipped with 10 electrodes at a sampling rate of 250 Hz (8 active and 2 reference electrodes) driven by a PIC32MX250F128B microcontroller. The system was used in combination with the OpenBCI, 19-channel EEG CAP (medium size) fitted with sintered Ag/AgCl wet electrodes, chosen due to their low impedance and stable signal acquisition properties [22], ensuring high-quality data capture over repeated sessions.

Electrode placement followed the internationally recognized 10–20 system [23] using the following channels: Fp1, Fp2, C3, C5, T5, T6, O1, and O2 targeting key frontal, parietal, temporal, and occipital regions (S1C Fig). To ensure consistent signal quality, a low-impedance electrode gel (OpenBCI 380130) was applied via a blunt-tipped syringe, and impedance was kept under 5 kΩ for all channels during the recordings.

The EEG signals were referenced to the right earlobe to establish a stable baseline. Additionally, the EEG system's bias electrode was connected to the subject's left wrist to minimize environmental noise and enhance the signal-to-noise ratio. This configuration allowed the differential amplifier to effectively reject common-mode noise by accounting for the bias voltage ($V_b$) in signal processing. The amplified EEG signal was obtained using the equation (I):

$$V_{\text{EEG, amplified}} = G \cdot [(V_{\text{EEG}} - V_b) - (V_{\text{reference}} - V_b)] \tag{I}$$

where, $V_{EEG}$ represents the raw EEG signal, $V_{reference}$ is the reference signal from the right earlobe, $V_b$ is the bias voltage, and G denotes the gain of the amplifier. Integrating the bias voltage into both signal paths, ensured a cleaner and more accurate representation of the neural activity.

To enhance participant comfort and reduce artifacts, data were transmitted wirelessly via the Cyton board's built-in radio module (over Bluetooth) to an Apple MacBook Pro computer with an M2 processor and 16 GB of RAM running the OpenBCI GUI software (version 6.0.0-beta.1).

The acquisition board was powered by a rechargeable lithium battery, ensuring a stable power supply throughout each session.

After each recording, the EEG cap was carefully maintained and cleaned following the manufacturer's protocol.

## 2.5 Signal processing and analysis

*EEG Data Preprocessing.* EEG data preprocessing was performed in MATLAB (MathWorks Inc., Natick, MA) and the EEGLAB toolbox [24]. Raw EEG signals were notch-filtered at 50 Hz to eliminate power line noise, followed by a band-pass filtering (0.5–80 Hz, Butterworth 2nd order filter, applied bidirectional) to isolate relevant EEG frequencies. Signals were then average referenced across all electrodes. Then, using the synchronization signal from the photoresistor, each recording was segmented into three periods: baseline or no light (5 minutes), constant light (5 minutes), and stimulus light (20 minutes). Noisy and artifact-heavy segments were manually inspected and removed, and Independent Component Analysis (ICA) was applied to remove ocular and muscle movement artifacts [25]. Subsequent analyses were conducted using the MATLAB EEGLAB toolbox (MATLAB R2024b, and EEGLab v2024.2.1). The preprocessing and analysis scripts are available at https://github.com/AminAlam/HVS.

*Fast Fourier Transform (FFT).* Power spectral density (PSD) estimates were obtained using FFT. The FFT algorithm computed the discrete Fourier Transform (DFT) of a sequence, which transforms discrete time-domain signals into their frequency-domain representation [26].

The DFT of a discrete signal $x[n]$ of length N is defined as in equation (II):

$$X[k] = \sum_{n=0}^{N-1} x[n]e^{-j\frac{2\pi}{N}kn}, \; k = 0, 1, \ldots, N-1$$

(II)

where $X[k]$ is the complex amplitude of the $k_{th}$ frequency component, $x[n]$ is the time-domain signal, $j$ is the imaginary unit, and $N$ is the number of samples. By applying the FFT, we were able to identify spectral components associated with the 60 Hz band light stimulation and assess changes in EEG power spectra across different experimental conditions.

*Topoplots.* Normalized power spectral density (PSD) values at 59.5–60 Hz were computed relative to the 2–80 Hz total power and further normalized to baseline (no light) condition. The normalized values were visualized using topoplots, allowing assessment of spatial distribution of 60 Hz signal changes across the scalp.

*Short-Time Fourier Transform (STFT).* To assess temporal dynamics of neural entrainment, STFT was applied using a Hamming window of 0.5 second length and 0.25 second overlap between windows. The STFT provides a way to analyze the frequency content of non-stationary signals over time by applying the Fourier Transform to short, overlapping segments of the signal [27].

The STFT of a discrete-time signal $x[n]$ is defined as in equation (III):

$$X(m, \omega) = \sum_{n=-\infty}^{\infty} x[n]w[n-m]e^{-j\omega n}$$

(III)

where $x[n]$ is the input signal, $w[n]$ is the window function (in this case, a Hamming window), $m$ is the time index corresponding to the center of the window, $\omega$ is the angular frequency, $j$ is the imaginary unit. This method enabled visualization of changes in power over time within each stimulation session, capturing transient or sustained changes in spectral power related to the light stimulation.

*Phase-Locking Value (PLV)*. Phase synchronization between EEG channels at 60 Hz was quantified using the PLV. EEG signals were filtered around 59.5–60.5 Hz using a 2nd order Butterworth filter applied bidirectionally, ensuring zero-phase distortion. The instantaneous phase of each signal was extracted using the Hilbert transform. The PLV between each pair of channels was computed using the formula [28] shown in equation (IV):

$$\text{PLV}_{ij} = \left| \frac{1}{N} \sum n = 1^N e^{j[\varphi_i(n) - \varphi_j(n)]} \right|$$

(IV)

where $\varphi_i(n)$ and $\varphi_j(n)$ are the instantaneous phases of signals $i$ and $j$ at time point $n$, $N$ is the number of time points, and $j$ is the imaginary unit.

The PLV for all light conditions was normalized against the PLV of the no light condition to quantify the relative increase in PLV compared to the baseline.

*Entrainment power over time.* To assess the temporal stability of 60 Hz entrainment during stimulation, the light stimulation step (step III) of each recording was divided into consecutive 4-minute bins. For each bin, the average normalized power at 60 Hz was calculated per electrode and plotted (each point on the plot represents the mean power from a single electrode within a given time bin). This approach allows visualization of how entrainment power evolves over time across both control and active conditions.

*Topographic Analysis.* To evaluate the spatial distribution and consistency of 60 Hz entrainment power and phase-locking value (PLV), electrodes were grouped into four anatomical brain regions: Frontal (Fp1, Fp2), Central (C3, C4), Temporal (T5, T6), and Occipital (O1, O2). Statistical comparisons of 60 Hz entrainment power were performed between the active and sham groups for each brain region. Additionally, PLV analyses were conducted across all inter-regional electrode pairings to compare phase synchrony patterns between the two groups.

## 2.6 Biomarkers and tolerability

*Saliva collection.* Saliva samples were obtained from participants on designated days (day 1, 5 and 19). Saliva samples were collected in 2 ml tubes (SARSTEDT, Cat. n. 72.695.500) following light stimulation, with the exact time of each collection carefully recorded. Participants were instructed to abstain from eating for a minimum of 1 hour before saliva collection and to avoid dairy products, caffeinated drinks, or acidic beverages within 15 minutes of sampling. Immediately following the collection, a Protease Inhibitor Cocktail (Sigma, Cat. n. #P2714) was added to the saliva at a concentration of 1 μL per 1 mL *(v/v)* of whole saliva, prepared according to the manufacturer's instructions. Samples were then centrifuged at 2,000 g for 2 minutes to remove large debris. The supernatant was aliquoted into 2 ml tubes, labeled with the participant ID code and date. Finally, samples were flash-frozen and stored at −70°C for subsequent analysis.

*Cortisol and CRP analysis.* Salivary cortisol and C-reactive protein (CRP) levels were measured using a competitive cortisol ELISA (Invitrogen, Cat. n. #EIAHCOR) and a quantitative CRP sandwich ELISA (Abcam, Cat. n. #ab108826). As a quality control step, samples with high viscosity, excessive debris, or mucus contamination were excluded from the analysis. Highly viscous saliva samples, which can be driven by dehydration, hormonal factors or bad dental status, were not compatible with pipetting [29]. To account for the strong diurnal variation of salivary cortisol levels, only samples collected within approximately the same time of the day (within 1h) were included in the cortisol analysis. Samples collected more than 1 hour apart were excluded due to reliability concerns. Ultimately, cortisol levels were analyzed in 9 participants (5 sham and 4 active), while CRP levels were analyzed in 10 participants (5 sham and 5 active).

The colorimetric reactions for both assays were measured using a Synergy H1 plate reader set to 450 nm, following the manufacturer's protocols. CRP and cortisol concentrations were calculated using AssayFit Pro version 5.3.2 online analysis software. Relative fold changes in cortisol levels over the course of the study were determined by normalizing each time point to baseline (day 1) cortisol levels.

**Safety, tolerability and blinding assessment.** To evaluate tolerability and gather general feedback on the stimulation, we designed a brief questionnaire and administered it to the participants at the end of the study. All participants (n = 14) completed the questionnaire. Adverse events were recorded daily based on spontaneous participants' self-reports throughout the study.

The questionnaire was in the form of three statements about the general experience, on which the participants were asked to answer. The questions and answer options were as follows:

Statement 1. "The stimulation was..." Response options: Pleasant or Unpleasant

Statement 2. "The stimulation was tolerable overall." Response options: rated on a scale from 0 (not at all) to 5 (absolutely).

Statement 3: "I would do the stimulation again." Response options: Rated on a scale from 0 (not at all) to 5 (absolutely).

### 2.7 Statistical analysis

*Biomarkers, demographic and tolerability.* Statistical analysis for biomarker, demographic and questionnaire data was performed using GraphPad (versions 5 and 10.4). The Shapiro-Wilk test was used to assess the normality of the data. Age and years of education did not show a normal distribution and were analyzed with a Wilcoxon rank-sum test. Cortisol levels followed a normal distribution and were analyzed using a two-way ANOVA to evaluate the interaction between group (active versus sham) and time. Since CRP levels did not meet the normality assumption, they were analyzed using the Kruskal-Wallis test. Ordinal questionnaire responses were evaluated using the Wilcoxon rank-sum test. Proportions (race, sex and blinding efficacy) were compared using Fisher's exact test. A significance threshold of $p < 0.05$ was applied to all tests.

*EEG entrainment power and PLV.* Statistical analyses were performed using MATLAB, with non-parametric tests used where violations of normality as determined by the Shapiro-Wilk test.

Inter-group comparisons (sham vs. active) for each stimulation condition and time point the Wilcoxon rank-sum test (also known as Mann-Whitney U test) was used.

Intra-group comparisons across multiple timepoints (days 1, 5, 19) within each group (sham or active) for a given light condition using the Kruskal-Wallis test.

Post-hoc pairwise comparisons: When the Kruskal-Wallis test indicated significant differences ($p < 0.05$), post-hoc pairwise comparisons were conducted using Dunn's test with Bonferroni correction.

## 3. Results

### 3.1 Participants and experimental setup used to study 60 Hz neural entrainment

The primary objective of this study was to assess whether 60 Hz light stimulation would elicit neural entrainment and synchrony in healthy volunteers.

To address this, we administered 60 Hz flickering light to a group of 8 volunteers (referred to as ´active´ group) and compared their response to a parallel group of 6 volunteers who received constant, non-flickering light (referred to as ´sham´ group). No statistically significant differences were observed across groups in terms of sex, age, education and race, indicating that the random allocation generated two well-balanced groups (**Table 1**). Two participants in the active group discontinued the study for personal reasons. Because of incomplete data, their EEG and biochemical

measurements were excluded from the data analysis. However, both individuals completed the final questionnaire, and their responses on subjective tolerability, blinding and side effects were included in the relevant analyses. The groups remained balanced also after the two dropouts (**Table 1**).

For the stimulation, we used a prototype wearable headset derived from safety glasses, lined with a strip of LEDs (**Fig 1A**, S1A Fig). The stimulation lasted for 3 weeks, with one session per day from Monday to Friday (**Fig 1C**). Weekend sessions were omitted to accommodate participant schedules and to ensure high compliance, a design commonly used in multi-week protocols of neurostimulation. Importantly, once a session was started, all participants completed it fully and there were no cases of interruptions or shortening of sessions.

Neural entrainment was investigated using an 8-channel EEG setup during the stimulation on days 1, 5 and 19. On EEG recording days, the stimulation followed a three-step experimental paradigm as illustrated in **Fig 1B** including no light, constant light and 60 Hz light stimulation. No light and constant light conditions served as controls: the no light condition was included to assess the participants' baseline brain activity without any external visual stimulation; the constant light condition allowed for the isolation of general effects of light exposure independent of the flickering characteristics of the stimulus light. This approach provided a comprehensive framework to evaluate the specificity of neural entrainment to the 60 Hz flickering light.

### 3.2 60 Hz light induces neural entrainment

We investigated the occurrence and characteristics of neural entrainment at 60 Hz using different EEG signal analysis techniques. First, we performed a Fast Fourier Transform (FFT) analysis to investigate the frequency component of the EEG signal and calculate the power spectral density (PSD). If entrainment occurs, we expect to observe a peak at the administered frequency.

Indeed, the FFT analysis of EEG data from participants in the active group demonstrated a clear and distinct peak in the 60 Hz band during the stimulus-modulated light exposure on day 1 (**Fig 1D**). This peak was absent in the baseline period and in all conditions for the sham group (**Fig 1D**, gray line, and S2A Fig).

The quantification of normalized entrainment power at 60 Hz band across individual channels (**Fig 1E**, F) confirms that the entrainment at 60 Hz band in the active group affected most channels in all subjects. The normalized power in the active group, averaged across channels per subject, was 2.8 on day 1 (SD 2.69), 2.4 on day 5 (SD 1.94) and 1.54 on day 19 (SD 0.61). The normalized power of individual channels ranged from 0.8 to 12.9 on day 1, from 0.7 to 9.3 on day 5 and from 0.7 to 3.6 on day 19.

On day 1, the mean of the distribution of normalized power at 60 Hz in the active group across channels from all subjects was significantly higher than that in the sham group (**Fig 1F**). As shown in **Fig 1F** (with shape code per each subject) and S2E Fig, while some variability was observed across subjects, all of them showed entrainment. Importantly, Time-frequency analysis using STFT revealed a consistent band of activity at 60 Hz throughout the 20-minute stimulus in the active group (**Fig 1G**). To investigate whether entrainment intensity would change over time, we binned the stimulation period in 4-minute intervals and compared temporal dynamics of the entrainment intensity across all individuals; this analysis showed no significant increase nor decrease of entrainment at 60 Hz over 20 minutes (S3 Fig). We did observe some variability in 60 Hz signals across the electrodes, with the strongest variability coming from the temporal ones (S4 Fig). Overall standard deviation is shown in S5 Fig.

This sustained 60 Hz activity was absent during the baseline period in the same participants and in the sham group (S2 Fig). The continuous presence of the 60 Hz band over time in the active group supports the effectiveness of the light stimulation in entraining neural oscillations at the desired frequency throughout the entire duration of the stimulation.

When analysing the effect of repeated stimulation over time, we observed a noticeable decrease in 60 Hz power on days 5 and 19 in the active group as compared to day 1, corresponding to a 14% and 45% decrease over day 1, respectively (**Fig 1D**, E, F). By day 19, the power in the active group was significantly reduced compared to day 1 in the same group (**Fig 1F**). However, the normalized power in the active group remained significant on each day as compared to the

sham group (**Fig 1F**). Overall, these results indicate that 60 Hz light stimulation facilitates neural entrainment, with the oscillation observed across multiple brain regions and a gradual decrease in power over time.

### 3.3  The 60 Hz light entrainment is widespread and synchronized

Having confirmed the occurrence of neural entrainment, we next assessed its spatial distribution. Topographical maps of EEG power in the 60 Hz band (**Fig 1E**, S2B Fig) showed a global increase in power across most electrodes on day 1 of stimulation in the active group. This increase was observed across frontal, central, temporal, and occipital regions, indicating widespread neural entrainment. We did not observe any topographically specific pattern of decline, with signal decreasing homogeneously across all channels over the 19 days (S4 Fig); however, the temporal channels showed the greatest variability across the 3 recording days.

Given the widespread pattern of brain activation, we performed phase-locking value (PLV) analysis to determine whether the 60 Hz neural entrainment observed across regions exhibited synchronization or consistent phase relationships. On day 1, strong phase synchronization was evident between most channel pairs in the active stimulation group (**Fig 2B**), suggesting coherent neural activity induced by the light stimulation.

Statistical analysis confirmed an increase in PLV in the active group compared to the sham (**Fig 2A**). Consistent with the findings on power of entrainment, also the PLV significantly decreased from day 1 till day 5 and day 19 (**Fig 2A**) in the active group. No changes in PLV were observed in the sham group (S6 Fig). Region-specific analysis revealed that the decline in PLV was not driven by any brain region but was homogeneous overall (S7 Fig). The observed reduction in PLV over time suggests a decline in neural synchrony at the stimulation frequency with repeated exposures. We must notice that, given the low number of subjects and the 8-channel EEG setup used, these results are preliminary and will need confirmation in a larger study with more resolution.

### 3.4  Effects on cortisol levels measured in saliva

After having explored the neural response to 60 Hz flickering light, we wondered whether this stimulation might elicit somatic responses that can be measured with biomarkers. Various NIBS, including TMS [30] and tDCS [31], have been shown to lower cortisol levels in healthy participants, suggesting a potential effect of neuromodulation on the neuroendocrine axis. As far as we know, this has not been investigated with light-induced neural entrainment, so we included this analysis in our experimental design. Salivary cortisol levels were presented as relative values, normalized to day 1 levels to account for sample variability as done by [32]. Although no statistically significant differences were observed between sham and active participants in response to 60 Hz stimulation at day 5 or day 19 (**Fig 3A**), active participants exhibited a notable trend toward reduced cortisol levels over the course of the stimulation. Future studies with a higher number of subjects might help elucidate whether 60 Hz neural entrainment influences cortisol levels.

### 3.5  Effects on CRP levels measured in saliva

In addition to the neuroendocrine axis, we wondered whether 60 Hz could have other somatic effects. We were particularly interested in potential immune responses, given the body of evidence indicating that both 40 Hz [15] and 60 Hz [19] can affect microglial phenotype in preclinical models. As a general marker of immune activation, we measured C-reactive protein (CRP) in saliva samples of participants. No significant differences were observed in the levels of CRP on day 1, 5 and 19, when comparing active and sham groups (**Fig 3B**). These findings suggest that 60 Hz stimulation, even over a three-week period, does not induce a detectable inflammatory response.

### 3.6  Tolerability

Neural entrainment using white light is generally safe and well tolerated by subjects with only minor side effects such as headaches and eye strain [33]. Our study was designed to minimize the discomfort experienced by subjects, by using an

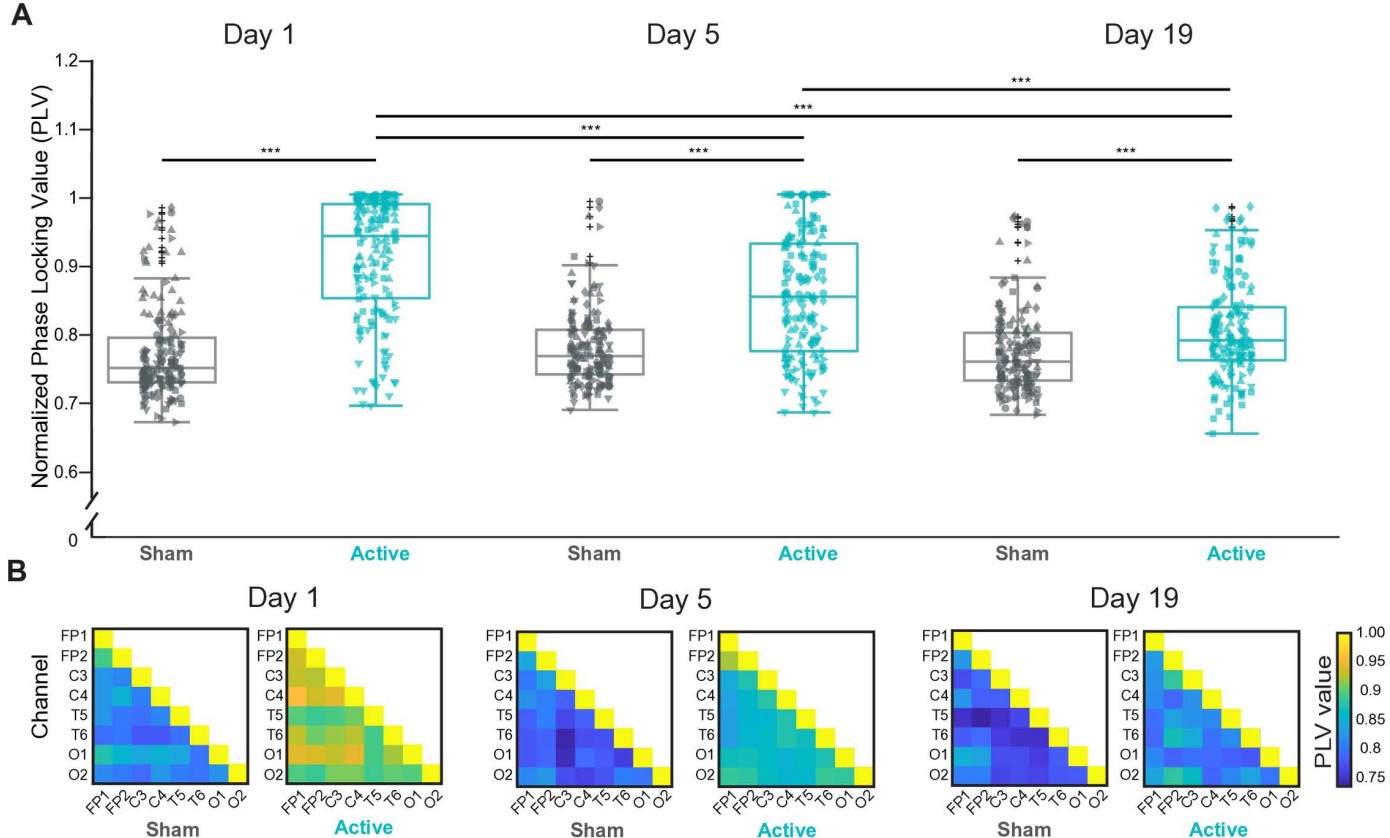

**Fig 2. Synchronization of brain activity during 60 Hz entrainment.** Phase-Locking Value (PLV) analysis was performed to evaluate whether 60 Hz neural entrainment in the active group led to synchronization or consistent phase relationships across brain regions. Each PLV matrix was normalized to its corresponding baseline PLV matrix (PLV matrix during No Light) to highlight stimulation-induced changes. **(A)** Significant differences in normalized PLV were observed across all channel pairs between the active and sham groups on days 1, 5, and 19. Additionally, within the active group, significant differences in normalized PLV were detected between day 1 vs. day 5 and day 1 vs. day 19, accounting for repeated measurements. Statistical significance for these inter-group comparisons was assessed using the Wilcoxon rank-sum test. The intra-group comparisons across days were evaluated using the Kruskal-Wallis test, followed by post-hoc pairwise comparisons performed using Dunn's test with Bonferroni correction. Normality of the data was assessed using the Shapiro-Wilk test, and non-parametric methods were employed due to deviations from normality. **(B)** PLV matrices for the active and sham groups across experimental days. Each element in the matrix represents the PLV for specific pairs of EEG channels, with diagonal elements showing a value of 1, indicating PLV between identical signals. * = p < 0.05, *** = p < 0.001. Detailed p-values are provided in S3 Table.

LED-generated light whose spectrum largely overlaps with daylight (S1B Fig), and at a very low intensity which subjects could adjust according to their preference (between 74–94 µW, measured at 2 cm distance from the LED strip). However, the tolerability of 60 Hz neural entrainment over extended periods has not been thoroughly studied. This is particularly relevant because our protocol induced an artificial and sustained 60 Hz activity pattern over three weeks which could potentially lead to discomfort.

To address this, we assessed the overall tolerability of the procedure through questionnaire, participant self-reports of side effects, and open-ended feedback. Questionnaire data, as well as side effects occurrence and dropouts, were compared statistically between the active and sham groups. The results of this analysis are shown in **Table 2** and **Fig 3C**.

Out of the 14 participants in the study, none found the stimulation, whether sham or active, unpleasant. The stimulation was rated as highly tolerable with an average score of 4.5 in the sham group and 4.8 in the active group (on a scale from 0 to 5). All participants indicated a willingness to undergo the procedure again, with an average score of 3.5 in the sham and 4.0 in the active group (on a scale from 0 to 5).

Side effects were recorded in both groups at similar rates and were considered minor.

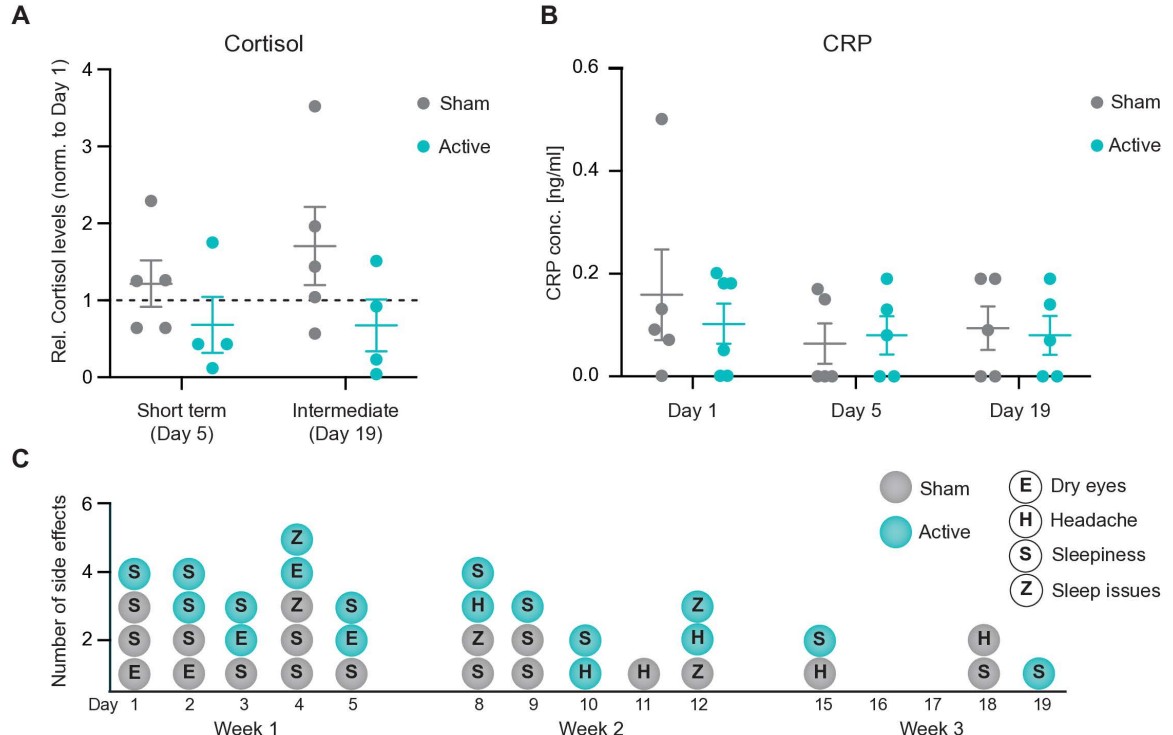

**Fig 3. Systemic effects of 60 Hz entrainment. (A-B)** Molecular marker analysis for stress and inflammatory responses in saliva following 60 Hz light stimulation. **(A)** Competitive ELISA quantification of salivary cortisol levels, normalized to baseline (day 1). Cortisol levels were measured after 1 week (day 5) of stimulation and after 3 weeks of intermediate stimulation (day 19) in the active (turquoise) and sham (gray) group (active: n = 4; sham: n = 5). Data are presented as scatter dot plots showing mean ± standard deviation (SD). Statistical significance was determined using two-way ANOVA with multiple comparisons. **(B)** Quantification of CRP levels in saliva, measured on day 1, day 5, and day 19 of the study. Participants received either active (turquoise) or sham (gray) stimulation (active: n = 5; sham: n = 5). Data are presented as scatter dot plots showing mean ± standard deviation (SD). Statistical significance was determined using Kruskal-Wallis test multiple comparisons. **C)** Day by day record of side effects during the stimulation. Each dot represents a single side effect. No statistically significant differences were observed across groups (see S3 Table).

Consistent with findings from other neural entrainment studies using 40 Hz [18,33,34], the most common side effect was sleepiness, reported by 66.7% of the sham group and 62.5% of the active group, followed by dry eyes/eye strain. Other reported side effects were new onset headache and sleep disturbances; quantification is presented in **Table 2**.

Interestingly, the number of reported side effects declined over time in both groups (total side effects in week one: 9 in active and 11 in sham group; total side effects in week 2: 7 in active and 6 in sham group; total side effects in week 3: 2 in active and 3 in sham group; see **Fig 3C**) corresponding to a decrease of 78% and 73% in active and sham groups, respectively, by week 3 as compared to week 1. Overall, the stimulation appeared to be well tolerated and safe, with no unexpected adverse events recorded. A similar percentage of participants guessed correctly their allocation in both groups (66.7% in the sham group, 62.5% in the active group), not statistically different from chance.

## 4. Discussion

### 4.1 Technical considerations

Previous studies have shown that the human visual system responds particularly well to rhythmic stimulation in specific frequency bands. Steady-state visual evoked potentials (SSVEPs) are robustly evoked in the alpha range (around 10 Hz)

**Table 2. Safety and tolerability.**

| | Sham group (n=6) | Active group (n=8) | p-value |
|---|---|---|---|
| Found the stimulation unpleasant, no (%) | 0 (0) | 0 (0) | n.a. |
| `The stimulation was overall tolerable` average score in a scale from 0 to 5±SD | 4.5±0.55 | 4.8±0.41 | 0.95 |
| `I would do the stimulation again` average score in a scale from 0 to 5±SD | 3.5±1.97 | 4±0.89 | 0.54 |
| Side effects total, number of events (% of subjects) | 20 (83.3) | 18 (87.5) | 1 |
| New onset headache | 3 (16.7) | 3 (25) | 1 |
| Dry eyes | 2 (33.3) | 3 (37.5) | 1 |
| Sleep disturbances | 4 (16.7) | 2 (12.5) | 1 |
| Sleepiness | 11 (66.7) | 10 (62.5) | 1 |
| No side effect | 1 (16.7) | 1 (12.5) | 1 |
| Drop outs, no (%) | 0 (0) | 2 (25) | 0.47 |
| Guessed correctly what stimulation they received, no (%) | 4 (66.7) | 5 (62.5) | 1 |

The Table shows the analysis of data collected from questionnaires (at the end of the study) and self-reports (throughout the study) to assess tolerability and safety of 60 Hz stimulation (active) as compared to stimulation with constant light (sham). Participants rated statements on a scale from 0 ('not at all') to 5 ('absolutely'). Both light stimulation conditions, sham and active, were well tolerated and induced only minor side effects. Discrete data were analyzed using the Mann-Whitney-Wilcoxon U test, while all other data were analyzed with Fisher's exact test. No statistically significant differences were observed between the sham and active groups.

and gamma (30–80 Hz) bands, although individual response profiles can vary [35] (reviewed in [36]). Within the gamma range, 40 Hz is the most extensively studied frequency, especially in relation to its therapeutic potential and its role in cognitive and sensory processing. However, entrainment has also been observed at higher gamma frequencies, such as 60 Hz [37,38].

While numerous studies have examined the effects of 40 Hz neural entrainment, the impact of 60 Hz entrainment remains relatively underexplored. High gamma oscillations, including those at 60 Hz, are functionally associated with memory [39] and their modulation with auditory stimulation [20] and tACS [21] have been shown to impact various high cognitive functions. In addition, such oscillations have been implicated in various neurological and psychiatric disorders [13]. Preclinical studies have demonstrated that 60 Hz visual stimulation can drive neuroplastic effects, such as perineuronal net remodeling [19], suggesting a potential therapeutic application.

One recent study [38] showed successful and widespread entrainment with acute 60 Hz visual stimulation, but the effect of prolonged stimulation has not been tested. A few studies have used transcranial alternating current stimulation (tACS) at 60 Hz in healthy volunteers and showed entrainment [40], but the effect of visual stimulation in this frequency has not been tested. The effects on somatic response of subjects and well-being have not been systematically recorded in most studies.

The scarcity of studies on 60 Hz entrainment might be due to inherent technical challenges in working with this specific frequency. For example, in North America, a 60 Hz EEG signal overlaps with electrical noise (voltage of 110 V and a frequency of 60 Hz), making it practically challenging to analyze the 60 Hz component and frequencies close to it. Being based in Europe, we worked with a voltage of 220 V at 50 Hz; therefore, the electrical noise could be easily filtered out and did not interfere with our analysis.

Another technical aspect that makes this type of work difficult is the interference between the electrical noise of the flickering LEDs mounted on the mask and the EEG recording. To overcome this issue, we have encased the LED strip in a Faraday cage. This resulted in elimination of the electrical noise as measured with multifield-EMF (data not shown). The fact that the 60 Hz entrainment signal in EEG declines over time in each subject, while the LED light was always active

throughout the 3 weeks with stable light intensity, demonstrates that we were successful in removing noise and that the observed signal is driven by brain activity.

The flickering light stimulation presented here is very different from that of older Cathode Ray Tube (CRT) monitors. CRTs refreshed the image by redrawing pixels, with phosphor glow persisting between frames, whereas research setups such as ours fully turn the light on and off in a precise, uniform manner. Additionally, CRT flicker was inconsistent (due to interlacing, phosphor decay, and screen refresh timing), whereas research flicker is synchronized across the entire visual field for a consistent neural response. It has been shown [41] that CRT displays with high refresh rates can induce some neural responses, particularly when a blank white screen is used, with the effect becoming stronger when very high-contrast stimuli are present. However, typical activities like watching a movie or browsing the web with normal contrast on an older CRT screen is unlikely to lead to significant neural entrainment.

### 4.2  60 Hz neural entrainment power

To our knowledge, this is the first study to combine detailed EEG signal analysis over a three-week period with assessments of somatic responses and tolerability in healthy volunteers receiving light-induced neural entrainment. The presence of the distinct 60 Hz peak in the PSD of active group participants confirmed that our intermittent light stimulation effectively entrained neural activity at the stimulation frequency. This finding aligns with previous studies demonstrating frequency-specific entrainment using acute stimulation with visual stimuli [42], as well as rhythmic sensory stimulation, particularly at gamma frequencies such as 40 Hz [11,35].

Further EEG analysis revealed entrainment effects in the active group participants and across the different EEG channels, though with noticeable inter-individual variability. The normalized power of individual channels ranged from 0.8 to 12.9 on day 1, 0.7 to 9.3 on day 5, and 0.7 to 3.6 on day 19. As shown in **Fig 1F**, most channels exhibited entrainment. Subject-specific responses, presented in **Fig 1F** and S2E Fig, revealed considerable variability across individuals; however, a 60 Hz entrainment was observed across all subjects. The STFT analysis (**Fig 1G**, S3 Fig) further demonstrated that entrainment was sustained over the 20-minute stimulation period for each individual.

### 4.3  60 Hz neural entrainment topographic distribution

Next, we investigated the topographical distribution of the entrainment. While our 8-channel setup lacked granular spatial resolution, we nevertheless had the opportunity to examine signal distribution across broadly-defined brain areas (occipital, temporal, central, and frontal regions) using topoplots and channel-specific analysis (respectively: O1 and O2; T5 and T6; C3 and C4; Fp1 and Fp2). Interestingly, we did not observe a focal neural activation limited to the occipital region (the primary target of visual stimulation) but rather an activity that spreads to the rest of the cortex, including the temporal cortex, reaching the frontal cortex. Similar spreading has been observed with other entrainment modalities such as 40 Hz light [34] and tES [43]. Such a diffuse pattern of brain activity could be driven by synaptic connections or traveling cortical waves. Indeed, gamma entrainment has been shown to induce traveling waves connecting the occipital/parietal regions to the frontal region of the cortex through the temporal region [44].

In addition, we found that the brain activation pattern is not random but is highly synchronized, as revealed by our PLV analysis. This finding is consistent with reports that flickering simulations increase phase connectivity between brain regions [45]. While the mechanism underlying such connectivity effect is not known, it is likely to involve functional neuroplastic changes. More studies are needed to elucidate this aspect.

### 4.4  Effect of repeated stimulations over time

Our experimental design allowed us to study in detail the changes in entrainment power comparing pre-stimulation and during stimulation conditions. We must note that post-stimulation effects or the potential persistence of entrainment were not explored in this experiment and would be the subject of subsequent studies.

In our experiment we have observed a change in the entrainment strength over time, namely after 5 and 15 days of stimulation (corresponding to day 5 and day 19). Our results indicate that the response is highest on day 1 during the very first stimulation, still present at day 5 but lower, and further reduced at day 19. For instance, the average normalized power across all channels of all subjects went from 2.8 on day 1 (SD 2.69) to 2.4 on day 5 (SD 1.94) and 1.54 on day 19 (SD 0.61). The change occurred over days and not during any of the individual stimulation sessions (S3 Fig). All features of the 60 Hz entrainment that we analyzed (the normalized entrainment power (Fig 1F), the topographic distribution (Fig 1E) and the PLV analysis (Fig 2A,B) showed a similar pattern. The decline in the entrainment and synchrony was uncorrelated with the level of brightness of the LEDs chosen by participants throughout the study, which stayed relatively stable over time with no reduction over the days (S1D, E Fig).

Interestingly, the change in entrainment and synchronization did not follow a specific topographic pattern but was observed across brain regions homogeneously (S4 Fig).

Such a reduction over time might be explained by a neural adaptation to the repeated stimulations. Similar adaptation has been previously described with repeated acoustic stimulations [46] as well as with 40 Hz combined visual and auditory entrainment [18]. The adaptation is likely related to homeostatic plasticity [47]. Briefly, repeated stimulations will first stimulate Hebbian plasticity and the formation of new connections (quick response in minutes/hours); this is followed, at a slower timescale (days), by homeostatic plasticity, which removes connections to ensure stability [48]. It is intriguing to observe that, in animal experiments, 60 Hz was shown by our group to promote juvenile-like neuroplasticity via remodeling of the PNN [19]. The hypothesis that 60 Hz flickering light induces functional synaptic changes via PNN remodeling in healthy volunteers remains to be investigated in larger studies.

## 4.5 Somatic response and tolerability

Evidence indicates that other NIBS can modulate the neuroendocrine axis [30,31] and can have an impact on microglia, the resident immune cells of the brain [15,19]. We were therefore interested in studying not just the brain response to 60 Hz flickering light, but also the somatic response. Importantly, we show that 60 Hz in healthy humans does not elicit a strong immune response, as CRP levels appear not statistically different across groups (Fig 3B). Similarly, the strong brain activity response we observed was not accompanied by any sign of stress, as measured by cortisol levels (Fig 3A); if anything, we observed a trend towards reduction. Considering the small sample size, the effect of 60 Hz on cortisol might deserve further investigation in a larger experiment.

Confirming the indication coming from the cortisol measurements, 60 Hz stimulation was overall well tolerated by the participants (Fig 3C). Based on the literature, mostly on 40 Hz stimulations [17,18,27], we expected only minor side effects. The vast majority of participants reported mild side effects, including eye strain, and sleepiness (see full list in Table 2). Notably, these effects were reported in both the sham and active groups at similar rates. As the side effects were minor and did not require medical attention, they are most likely attributable to general light exposure rather than the 60 Hz flickering specifically. The responses the participants gave to our questionnaire confirmed that the stimulation, both active and sham, was well tolerated, and all participants indicated that they would be willing to undergo the stimulation again. The two dropouts occurred due to personal and stimulation-unrelated reasons. No participant requested to interrupt or shorten a stimulation session at any point. Since this was a particularly young and healthy research cohort, it remains to be assessed whether the 60 Hz effect is the same in a more heterogeneous population. Nonetheless, the high tolerability observed across the stimulation sessions supports the interpretation that the EEG changes were not confounded by stress or discomfort caused by the stimulation itself.

## 4.6 Limitations

We identified the following limitations in our study: First, the small sample size, particularly in the active group (n=6), limits statistical power and the generalizability of our findings. The sample size was approved by the Ethics Commission as appropriate for an initial feasibility study and is in line with other EEG and NIBS studies with sample sizes lower than 10

per group [49,50]. However, due to the small sample size, we were unable to explore potential influences of sex, gender, age, and time of day on the neural response to 60 Hz flickering light. In summary, these open points remain important areas for future research. Nevertheless, despite the low number of subjects, our findings suggest neural entrainment after 60 Hz light stimulation and its evolution over time.

Second, while we did our best to ensure a balance for sex and age, the participants of this study were all very young, highly educated, and mostly white. The neural entrainment following 60 Hz flickering light in a more diverse population remains to be established.

Third, the use of an 8-channel EEG gave us low spatial resolution and did not allow us to study in detail the topographical distribution of the entrainment.

Fourth, the absence of a 40 Hz control condition limits the interpretability of the frequency-specific effects observed. While our primary aim was to investigate the feasibility and tolerability of 60 Hz flickering light, future studies should include direct frequency comparisons to better contextualize these findings.

## 5. Conclusions

In this study, externally induced neural entrainment with 60 Hz flickering light over 3 weeks was strong and synchronized across brain regions, it showed no major side effects and presented potential signs of neural habituation. Several FDA-approved drugs for psychiatric conditions, including SSRIs and ketamine, boost neuroplasticity [51,52], promote the remodeling of extracellular matrix [19], and induce gamma activity [53]. Externally induced 60 Hz entrainment might therefore be a new approach for modulating brain activity and inducing neuroplasticity, with implications for our basic understanding of brain physiology as well as treatment of psychiatric disorders. Since light is a non-invasive, easily implementable, amenable to at-home treatment, and relatively cheap methodology, this research area warrants further investigation.

## Supporting information

**S1 Table. Inclusion and Exclusion criteria.**
(DOCX)

**S2 Table. Manufacturer and catalogue number details of material used.**
(XLSX)

**S3 Table. Statistical analysis data.**
(XLSX)

**S1 Fig. Experimental setup to assess 60 Hz-induced brain entrainment. (A)** Photographs of the experimental setup on EEG days: subjects were seated on a chair during the stimulation (picture was taken of one of the authors as an example). **(B)** Graph showing the light spectrum of the LEDs ranging from 440 nm to 770 nm, similar to daylight wavelengths. **(C)** The figure illustrates the location of the 8 EEG electrodes utilized in the study, mapped according to the 10–20 electrode placement system. **(D)** Analysis of individual light intensities chosen by participants (mean ± SD) across all sessions (15 days – 3 weeks, 5 days a week) and their distribution, demonstrating stable intensity settings within the comfort range (74–94 µW).
(TIF)

**S2 Fig. Control conditions for baseline, I. No light and II. Constant light stimulation (A)** Scalp EEG power spectral density (PSD) averaged across all channels for participants in each group under (I.) No light and (II.) Constant light conditions. The gray bar indicates the 50 Hz line noise, which was notch-filtered. **(B)** Topographic maps showing normalized changes in 60 Hz PSD (relative to No light) averaged across participants of each group under (I.) No light and (II.)

Constant light conditions. (**C, D**) 60 Hz PSD values across all channels between the active and sham groups under (I.) No light and (II.) Constant light conditions on days 1, 5, and 19. The observed statistically significant difference between Constant light condition of Active group day 1 and Active group day 19 is driven primarily by the large sample size (n), rather than any meaningful biological variation, as the average normalized 60 Hz power across these two days remains comparable. Statistical significance for these inter-group comparisons was assessed using the Wilcoxon rank-sum test. Furthermore, within the active group, significant differences in normalized PSD were detected between day 1 vs. day 19, accounting for repeated measurements. These intra-group comparisons across days were evaluated using the Kruskal-Wallis test, followed by post-hoc pairwise comparisons performed using Dunn's test with Bonferroni correction. The normality of the data was assessed using the Shapiro-Wilk test, and non-parametric methods were applied due to deviations from normality. In this figure, different subjects are shape-coded. Only significant differences are indicated, with * indicating $p < 0.05$. Detailed p-values are provided in **S3 Table**. (**E**) The figure displays the average normalized PSD of all channels for each subject in both groups over days 1, 5, and 19, represented as individual lines. The thick line indicates the group average across all channels of all subjects. Data were recorded during light stimulation, with 60 Hz flickering light for the active group and constant light for the sham group.
(TIF)

**S3 Fig. Temporal dynamics of 60 Hz entrainment over 25-minute stimulation sessions.** Each dot represents the normalized power spectral density (PSD) at 60 Hz from an individual EEG channel of a subject, normalized relative to the "No light" baseline. The line shows the trajectory of the group median across time points during each session. The stimulation phase, corresponding to the "Stimulus Light" condition, spans 20 minutes and is divided into four-minute time bins (0–4, 4–8, 8–12, 12–16, and 16–20 minutes). Normalized 60 Hz PSD values from all channels are averaged for each subject and then grouped by experimental condition (active vs. sham) on Days 1, 5, and 19. The figure illustrates how 60 Hz entrainment evolves temporally across sessions, highlighting the stability of 60 Hz entrainment over a 20 minutes stimulation period.
(TIF)

**S4 Fig. Regional dynamics of 60 Hz power across frontal, temporal, central and occipital regions during flickering light stimulation. (A)** Box plots depicting normalized 60 Hz power spectral density (PSD) relative to no light condition, averaged over electrode clusters representing the frontal (Fp1, Fp2), temporal (T5, T6), central (C3, C4), and occipital (O1, O2) regions, in the active and sham groups, across days 1, 5, and 19. Significant differences are indicated with *$p < 0.05$, **$p < 0.01$, and ***$p < 0.001$. Detailed p-values are reported in **S3 Table**.
(TIF)

**S5 Fig. Short-time Fourier transform (STFT) analysis of 60 Hz entrainment.** (A) Average STFT time-frequency plots for a representative sham participant (averaged across all electrodes) illustrate the absence of a visible 60 Hz signal during constant light exposure. (B) The standard deviation (SD) of the STFT power at 60 Hz for the same representative sham participant confirms low variability and no entrainment at the stimulation frequency. (C) Standard deviation (SD) of STFT power at 60 Hz across all electrodes for a representative active group participant on day 1. The green arrow indicates variability in the 60 Hz entrainment due to the variability in the level of entrainment in different channels.
(TIF)

**S6 Fig. Assessment of phase synchronisation of brain activity during 60 Hz entrainment during no light and constant light stimulus.** (**A**) Significant differences in normalized PLV (normalized to PLV matrix of No Light condition) were observed across all channel pairs between the active and sham groups under Constant light conditions on days 1. Inter-group comparisons were assessed using the Wilcoxon rank-sum test. Significant differences were observed in PLV measurements for the active group between day 1 and days 5 and 19. Intra-group comparisons across days were evaluated

using the Kruskal-Wallis test, followed by post-hoc pairwise comparisons performed using Dunn's test with Bonferroni correction. The statistically significant differences in PLV observed under constant light conditions are likely attributable to the large number of channel pair comparisons (n) rather than meaningful biological variation, since the average PLV values across groups and days remain comparable. Normality of the data was assessed using the Shapiro-Wilk test, and non-parametric methods were employed due to deviations from normality. (B) PLV matrices for the active and sham groups across experimental days under (I.) No light and (II.) Constant light conditions. Each element in the matrix represents the PLV for specific pairs of EEG channels, with diagonal elements showing a value of 1, indicating PLV between identical signals. Significant differences are indicated, with ** indicating $p < 0.01$ and *** indicating $p < 0.001$. Detailed p-values are provided in **S3 Table**.
(TIF)

**S7 Fig. Regional analysis of phase synchronization of brain activity during 60 Hz light stimulation.** Box blots depicting normalized phase-locking value (PLV) at 60 Hz between the frontal, temporal, central, and occipital regions on days 1, 5, and 19. This highlights the spatial patterns of phase synchronization during repeated stimulation sessions. (**A**) shows PLV values during constant light conditions, and (**B**) shows PLV values during stimulus light conditions (60 Hz flicker for the active group and constant light for the sham group). Statistical significance was assessed using the Wilcoxon rank-sum test. Normality of the data was assessed using the Shapiro-Wilk test, and non-parametric methods were employed due to deviations from normality. Significant differences are indicated with *$p < 0.05$, **$p < 0.01$, and ***$p < 0.001$. Detailed p-values are provided in **S3 Table**.
(TIF)

## Author contributions

**Conceptualization:** Maria Teresa Ferretti, MohammadAmin Alamalhoda, Francesca Giovanetti, Mark Caffrey, Jack O'Keeffe, Alessandro Venturino.

**Data curation:** MohammadAmin Alamalhoda, Friederike Leesch.

**Formal analysis:** Maria Teresa Ferretti, MohammadAmin Alamalhoda, Friederike Leesch.

**Funding acquisition:** Mark Caffrey, Jack O'Keeffe, Alessandro Venturino.

**Investigation:** Maria Teresa Ferretti, MohammadAmin Alamalhoda, Friederike Leesch, Alessandro Venturino.

**Methodology:** Maria Teresa Ferretti, MohammadAmin Alamalhoda, Friederike Leesch, Alessandro Venturino.

**Project administration:** Francesca Giovanetti.

**Resources:** Mark Caffrey, Jack O'Keeffe.

**Software:** MohammadAmin Alamalhoda.

**Supervision:** Maria Teresa Ferretti, Alessandro Venturino.

**Validation:** MohammadAmin Alamalhoda, Friederike Leesch, Alessandro Venturino.

**Visualization:** MohammadAmin Alamalhoda, Friederike Leesch.

**Writing – original draft:** Maria Teresa Ferretti, MohammadAmin Alamalhoda, Friederike Leesch, Alessandro Venturino.

**Writing – review & editing:** Maria Teresa Ferretti, MohammadAmin Alamalhoda, Friederike Leesch, Francesca Giovanetti, Eoghan Dunne, Giuseppina Pilloni, Mark Caffrey, Jack O'Keeffe, Alessandro Venturino.

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
