## [Decision Letter · Decision Letter 0]

15 May 2025

PONE-D-25-10411Flickering white light stimulation at 60 Hz induces strong, widespread neural entrainment and synchrony in healthy subjectsPLOS ONE

Dear Dr. ferretti,

Thank you for submitting your manuscript to PLOS ONE. After careful consideration, we feel that it has merit but does not fully meet PLOS ONE’s publication criteria as it currently stands. Therefore, we invite you to submit a revised version of the manuscript that addresses the points raised during the review process.

Reviewers agree the topic is interesting but identify several critical shortcomings that preclude acceptance in its current form. First, the choice of sixty hertz is weakly justified because the established forty hertz control is absent and supporting citations are missing. Second, the sample size of six completers per arm is too small to sustain claims about adaptation or safety, so the authors must recruit more participants or temper their conclusions. Third, the methods are not described in enough detail; information on light-intensity calibration, full EEG preprocessing, questionnaire content, and statistical decision rules is required to rule out confounds such as day-to-day intensity drift. These gaps undermine the reported decline in entrainment because adaptation is not tracked daily, across weekends, or after the treatment ends. Electrode-level differences remain qualitative, and demographic balance after dropouts is unclear.  Addressing these points is essential before the work can be considered further..

We look forward to receiving your revised manuscript.

Kind regards,

Gennady S. Cymbalyuk, Ph.D.

Academic Editor

PLOS ONE

 [This study was funded by Syntropic Medical and supported by an Austria Wirtschaftsservice (AWS) grant (grant number P2414247 to Syntropic Medical).]. 

[This study was funded by Syntropic Medical and supported by an Austria Wirtschaftsservice (AWS) grant (grant number P2414247 to Syntropic Medical).

The authors      thank all the participants for their time and commitment to this study. Special thanks to Dr. Verena Seiboth, Mrs. Verena Schmied, and Dr. Katalin Szigeti for their valuable assistance, and to the Austrian Research Promotion Agency (FFG).]

 [This study was funded by Syntropic Medical and supported by an Austria Wirtschaftsservice (AWS) grant (grant number P2414247 to Syntropic Medical).]

Reviewers' comments:

Reviewer's Responses to Questions

**Comments to the Author**

1. Is the manuscript technically sound, and do the data support the conclusions?

Reviewer #1: No

Reviewer #2: Partly

2. Has the statistical analysis been performed appropriately and rigorously? 

Reviewer #1: Yes

Reviewer #2: No

3. Have the authors made all data underlying the findings in their manuscript fully available?

Reviewer #1: Yes

Reviewer #2: Yes

4. Is the manuscript presented in an intelligible fashion and written in standard English?

Reviewer #1: Yes

Reviewer #2: Yes

5. Review Comments to the Author

Reviewer #1: Major comments:

1. The choice of 60 Hz for stimulation requires clearer justification. Given the growing body of evidence supporting 40 Hz entrainment in modulation neural activity, a comparison between 40 Hz and 60 Hz stimulation would provide a more informative perspective. The rationale for selecting 60 Hz should be explicitly discussed, particularly in relation to known effects.

2. The final sample size (n = 6 per group after dropouts) is small, particularly for EEG-based study on younger adults. Furthermore, the observed subject variability in entrainment strength across the 3 EEG sessions makes drawing any strong conclusions about the habituation effect difficult.

3. It is unclear whether light intensity was adjusted on each day and how this was accounted for in the analyses. If intensity varied across days or between participants, this could have confounded the results. As stimulus intensity can influence the response, it is important to confirm that the total stimulus power was comparable between sessions and conditions. Without this control, it is difficult to determine whether reduced entrainment strength is due to habituation or changes in stimulus intensity.

4. The Methods section lacks a description of EEG preprocessing and analysis steps, which are critical for interpreting the EEG findings. Additionally, the questionnaires used for assessing tolerability and other outcomes are not described. These methodological omissions should be addressed to ensure reproducibility and transparency.

5. The manuscript would benefit from additional analysis of the temporal dynamics of the observed effects (comparing pre-, during, and post-stimulation). This would provide additional insight into whether such stimulation has transient or longer-lasting effects.

6. Demographic comparisons appear to have been conducted on the full sample, but two participants in the Active group were excluded from EEG and biomarker analysis. It would be helpful to clarify whether the demographic balance remained after these dropouts, especially since the final analysis excluded their data.

7. The authors mention that statistical test selection was based on the outcome of normality tests. However, repeated measures were analyzed with two-way ANOVA, which assumes normality. It should be clarified whether ANOVA was applied only to variables that passed normality checks, or if a non-parametric alternative was considered when assumptions were violated.

8. The Discussion references “quality-of-life” assessments, but these are not clearly defined or discussed elsewhere in the manuscript. It is important to distinguish between assessing tolerability/safety and evaluating quality of life, as these represent different outcomes.

Minor comments:

1. Several sentences should be revised. E.g., “Additional infos on light stimulation device and the mini-Faraday cage in Supplementary Methods”, “Additional infos and EEG data analysis in Supplementary Methods”.

2. Some sections include excessive technical detail (e.g. manufacturer part numbers, catalog numbers) that could be moved to the Supplementary Materials to improve readability and flow.

Reviewer #2: The authors investigated the effect of 20-minute, 60 Hz light flickering stimulation applied on a daily basis for the first 5 days during 3 consecutive weeks on EEG activity measured with an 8-channel system in 8(6) healthy young volunteers versus 6 controls. They also collected saliva samples to evaluate CRP and cortisol levels immediately after light stimulation, which were not elevated in either of the two groups. There was no task or behavioural assessment involved, other than general questions about well-being, which did not show any systematic difference between the active and the sham groups. Participants receiving flicker stimulation presented 60 Hz entrainment on all 8 electrodes, as revealed by power spectral density measures and overall synchrony between the signals for all electrodes studied using pairwise phase-locking values. By the 5th day of light treatment, there was a significant decline in the 60 Hz-related power on the electrodes, and an apparently even stronger decline by the 19th day. The authors conclude that this decline may reflect adaptation and/or plasticity, that the protocol has no harmful effects and that it might thus be suitable for clinical treatment.

General comments

1. The choice of 60 Hz is not sufficiently justified. One of the reasons the authors cite is that this frequency was not explored in humans before, but they fail to cite other studies using 60 Hz sensory stimulation who also compare effects of other frequencies (e.g., Manippa et al., 2024) and their effects on task performance. This set of studies suggested a positive effect of 60 Hz auditory stimulation on executive functions such as cognitive control and attention (not memory).

2. At least one lower frequency should have been tested as a control. In the cited animal study (Venturino et al., 2021), 60 Hz—compared to 40 Hz—had a specific effect on PNN/microglia, which could have been directly contrasted.

3. The number of participants who completed the EEG study is too low, with only 6 in the active group. Since the protcol is not harmful, this low number might be reconsidered. Maybe, because of that, the data also seem under-analysed with regard to effects within individuals over time, and analysis of differences between electrodes. Given that there are reports that gamma stimulation does not even reach the hippocampus that point might be important to adress.

4. The protocol does not appear fully consistent. What was the reason for not stimulating on weekends? One would expect daily stimulation to observe adaptation dynamics more precisely.

5. EEG recordings should have been conducted on consecutive days during at least one week to investigate the timeline of entrainment decline. For example, was there any recovery over the weekend, or was the decline sharper during consecutive days?

6. It remains unclear how long the adaptation effect lasted after the flicker stimulation ended. One could have included a follow-up.

7. There are references missing on the impact of sensory gamma stimulation on the glymphatic system.

Detailed comments

1. Figure 1G: which electrode is shown? Please show at least two different ones.

2. Although the overall analysis suggests entrainment and adaptation across all electrodes, there seem to be considerable differences between them as suggested by the images. These differences should be quantified (for example, in Figure 2A, active group day 1).

3. A comment should be made on why the frontal or temporal electrodes seem to adapt more than the occipital ones, which are closer to the flickering visual input.

4. A discussion and citations should be included on which stimulation frequencies are able to induce entrainment in the visual system.

5. How much decline in 60 Hz entrainment occurred during a single 20-minute session after 5 and 19 days?

6. PLOS authors have the option to publish the peer review history of their article (what does this mean? ). If published, this will include your full peer review and any attached files.

**Do you want your identity to be public for this peer review?** For information about this choice, including consent withdrawal, please see our Privacy Policy .

Reviewer #1: No

Reviewer #2: No

---

## [Author Response · Author response to Decision Letter 1]

14 Jul 2025

Reviewer #1

Major comments:

1. The choice of 60 Hz for stimulation requires clearer justification. Given the growing body of evidence supporting 40 Hz entrainment in modulation neural activity, a comparison between 40 Hz and 60 Hz stimulation would provide a more informative perspective. The rationale for selecting 60 Hz should be explicitly discussed, particularly in relation to known effects.

We thank the Reviewer for this important comment. In the revised introduction, we clarified and expanded the rationale for selecting 60 Hz stimulation and included relevant citations demonstrating its emerging significance. Specifically, we now reference recent studies using 60 Hz transcranial alternating current stimulation (tACS) and auditory stimulation in humans, including those by:

Manippa et al. (2024), who reported reduced intrusion errors and enhanced cognitive control in healthy adults following 60 Hz auditory stimulation,

Nomura et al. (2019), who demonstrated that 60 Hz tACS over the dorsolateral prefrontal cortex (DLPFC) enhances declarative long-term memory (LTM).

These additions provide context regarding the functional relevance of 60 Hz gamma-band activity in executive and memory-related processes and address the concern that the original text lacked supporting human data.

Further, we emphasize that while 60 Hz has been tested in auditory and electrical protocols, its use in visual entrainment, especially over multiple weeks, remains largely unaddressed. Our study aims to explore this area. This point has been explicitly stated in the revised introduction (page 5 lines 97-105).

We agree that a lot is known on the effects of 40 Hz and a direct comparison with 60 Hz would be interesting to perform. Our goal for the present study was to generate a knowledge base with a full characterization of 60 Hz light stimulation over time – something that is missing in the literature. As a subsequent study, a direct comparison with 40 Hz would be highly relevant. To acknowledge this, in the Limitations section, we added a paragraph acknowledging the limitations of our study, such as the absence of 40 Hz for direct comparison of effects, as well as the broader absence of comparable human data on the effects of different frequency visual stimulation. By clearly outlining these gaps and open questions, we hope to facilitate future research on frequency-specific neuromodulation.

2. The final sample size (n = 6 per group after dropouts) is small, particularly for EEG-based study on younger adults. Furthermore, the observed subject variability in entrainment strength across the 3 EEG sessions makes drawing any strong conclusions about the habituation effect difficult.

We acknowledge the Reviewer's concerns about the small sample size and its implications for statistical power. Our study was designed (and approved by the Ethics Commission of Lower Austria) to be an initial, pilot investigation to evaluate the feasibility, safety and neurological effects of 60 Hz flickering light stimulation over an extended period, which was never done before in the literature. Such a small sample size has been previously used in some EEG studies, such as Sharpe et al., 2020 and Mangia et al., PLOS ONE 2014 (now quoted).

We also agree with the Reviewer on the high variability in entrainment strength, which limits our ability to make strong conclusions about habituation. In response, we have revised the manuscript to temper these claims, emphasising the exploratory nature of our findings and the necessity for larger-scale studies to validate and extend our results. These clarifications have been incorporated into the abstract (page 2, line 40-43 and 48-49) and discussion section (pages 31- 35, sections 4.2 to 4.5, and especially 4.6 limitations pages 36-37). We have also modified the title accordingly, which now reads ´Exploring Neural Entrainment and Synchrony in Response to Repeated 60 Hz Flickering White Light in Healthy Volunteers´.

3. It is unclear whether light intensity was adjusted on each day and how this was accounted for in the analyses. If intensity varied across days or between participants, this could have confounded the results. As stimulus intensity can influence the response, it is important to confirm that the total stimulus power was comparable between sessions and conditions. Without this control, it is difficult to determine whether reduced entrainment strength is due to habituation or changes in stimulus intensity.

We thank the Reviewer for this valuable feedback, indeed this is an important point to clarify. The stimulation protocol was designed to ensure safety and comfort, allowing for individualised light intensity settings (within a given range) based on verbal feedback from participants. To assess whether the intensity chosen could influence the entrainment and habituation observed, we have now performed an additional analysis to explore the changes in intensity levels selected by participants across the days. This analysis (presented in SI 1Fig D-E) revealed that participants chose an average light intensity of approximately 80 μW (SD: 7 μW). Notably, intra-individual variation over the 15 stimulation days was minimal, with a maximum SD of 9 μW for any single participant, representing less than 11% of the mean intensity. No decline in light intensity was observed over the days of stimulation, suggesting that this is not a factor involved in the observed habituation. Given this consistency across subjects and days, and the absence of a reduction in intensity over time, we believe it is unlikely that small day-to-day variations in stimulus power explain the observed reduction in entrainment strength over time. This important point has now been clarified in both the Methods and Discussion sections.

4. The Methods section lacks a description of EEG preprocessing and analysis steps, which are critical for interpreting the EEG findings. Additionally, the questionnaires used for assessing tolerability and other outcomes are not described. These methodological omissions should be addressed to ensure reproducibility and transparency.

We thank the Reviewer for this valuable observations regarding the Materials and Methods section, giving us an opportunity to improve its readability and transparency. We agree that the choice to place some of the key information in Supplementary might not have been ideal, so we have significantly restructured the M&M section accordingly. First, we have incorporated the supplementary materials and methods section into the main text – this includes thorough details on EEG preprocessing (Page 11 lines 248-260) and tolerability and blinding questionnaires (page 16, lines 353-367). In addition, to increase readability, in the revised manuscript, we have thoroughly restructured several paragraphs, including the Signal Processing and Analysis section, and added clearly labeled subheadings for each step of the EEG pipeline: “EEG Preprocessing,” “FFT Analysis,” “Topoplots,” “STFT,” and “PLV Calculation.”. Also, addressing Minor Comment 2 below, we have moved the technical details (manufacturer´s names and catalogue numbers of device components) to a separate table (SI Table 2), to improve the flow.

5. The manuscript would benefit from additional analysis of the temporal dynamics of the observed effects (comparing pre-, during, and post-stimulation). This would provide additional insight into whether such stimulation has transient or longer-lasting effects.

We thank the Reviewer for this suggestion regarding a further exploration of our data to assess the temporal dynamics of neural entrainment.

Each EEG session included three consecutive recording phases: a five-minute baseline period (no light), a five-minute period of constant light exposure, and a 20-minute stimulation phase (60-Hz flickering light or constant light). This design enabled us to compare neural activity pre-stimulation and during stimulation within each session.

Following the Reviewer’s comment, we have performed an additional temporal analysis to characterize the dynamics of entrainment during stimulation with more granularity. Specifically, we have divided the 20 min stimulation period into five non-overlapping, 4 minute intervals and analyzed the normalized 60-Hz power within each segment. The results showed that entrainment strength remained stable throughout the 20-minute stimulation phase, across all three days (Day 1, Day 5, and Day 19). This suggests sustained entrainment, with no evidence of decay or rapid adaptation during the stimulation period. This new analysis is included as SI Fig 3.

However, we acknowledge that our protocol did not include EEG recordings after the stimulation phase. Consequently, we cannot directly assess post-stimulation effects or the potential persistence of entrainment. We now explicitly state this limitation in the Results section (page 33, lines 755-758). The Results paragraph now reads:

Our experimental design allowed us to study in detail the changes in entrainment power comparing pre-stimulation and during stimulation conditions. We must note that post-stimulation effects or the potential persistence of entrainment were not explored in this experiment and would be the subject of subsequent studies.

6. Demographic comparisons appear to have been conducted on the full sample, but two participants in the Active group were excluded from EEG and biomarker analysis. It would be helpful to clarify whether the demographic balance remained after these dropouts, especially since the final analysis excluded their data.

We thank the Reviewer for highlighting this important aspect. Although initially conducted on the full randomized sample (N = 14), we agree that it is important to assess demographic balance within the final analysis cohort (N = 12), which comprises only participants with complete EEG and biomarker data. A re-analysis of the data was therefore performed. As a result we found that the 2 drop outs did not substantially affect the group balance, since there were no statistically significant differences between the groups in the final analysis sample in terms of age, sex, race or years of education. Updated group statistics and comparisons are included now in Table 1 for transparency and a clarification has been added in the accompanying text and caption (page 18) as follows:

Two participants in the active group discontinued the study for personal reasons. Because of incomplete data, their EEG and biochemical measurements were excluded from the data analysis. However, both individuals completed the final questionnaire, and their responses on subjective tolerability, blinding and side effects were included in the relevant analyses. The groups remained balanced also after the two dropouts (Table 1).

7. The authors mention that statistical test selection was based on the outcome of normality tests. However, repeated measures were analyzed with two-way ANOVA, which assumes normality. It should be clarified whether ANOVA was applied only to variables that passed normality checks, or if a non-parametric alternative was considered when assumptions were violated.

We thank the Reviewer for pointing this to us. We are sorry that our text in the Statistic section was not clear in this regard. Indeed all data have been tested for normality and parametric tests have been used only when the normality was ensured. We have now edited the section (page 17) to clarify this, as follows:

´Statistical analysis for biomarker, demographic and questionnaire data were performed using GraphPad (versions 5 and 10.4) The Shapiro-Wilk test was used to assess the normality of the data. Age and years of education did not show a normal distribution and were analyzed with a Wilcoxon rank-sum test. Cortisol levels followed a normal distribution and were analyzed using a two-way ANOVA to evaluate the interaction between group (active versus sham) and time. Since CRP levels did not meet the normality assumption, they were analyzed using the Kruskal–Wallis test. Ordinal questionnaire responses were evaluated using the Wilcoxon rank-sum test. Proportions (race, sex and blinding efficacy) were compared using Fisher's exact test. A significance threshold of p < 0.05 was applied to all tests.´

8. The Discussion references “quality-of-life” assessments, but these are not clearly defined or discussed elsewhere in the manuscript. It is important to distinguish between assessing tolerability/safety and evaluating quality of life, as these represent different outcomes.

We thank the Reviewer for highlighting this distinction. We acknowledge that the term 'quality of life' was used inaccurately in the Discussion, as we did not administer any standardised instruments that would enable us to assess quality of life. In our study, we collected subjective reports related to tolerability and perceived well-being, including minor side effects, willingness to continue stimulation and overall participant impressions.

To address this, we have revised the relevant sentence in the Discussion section to reflect the scope of the data collected. The term 'quality of life' has been replaced with a more appropriate description of tolerability and subjective experience (page 35).

Reviewer 1 Minor comments:

1. Several sentences should be revised. E.g., “Additional infos on light stimulation device and the mini-Faraday cage in Supplementary Methods”, “Additional infos and EEG data analysis in Supplementary Methods”.

Thanks for this comment. We have revised the text to reflect the fact that the additional methods are now in the main manuscript text, and edited those sentences accordingly.

2. Some sections include excessive technical detail (e.g. manufacturer part numbers, catalog numbers) that could be moved to the Supplementary Materials to improve readability and flow.

We concur with the Reviewer and have moved the technical details (manufacture´s names and catalogue numbers for device components) to a separate Supplementary table (SI Table 2), to improve the flow and readability.

Reviewer #2

General comments

1. The choice of 60 Hz is not sufficiently justified. One of the reasons the authors cite is that this frequency was not explored in humans before, but they fail to cite other studies using 60 Hz sensory stimulation who also compare effects of other frequencies (e.g., Manippa et al., 2024) and their effects on task performance. This set of studies suggested a positive effect of 60 Hz auditory stimulation on executive functions such as cognitive control and attention (not memory).

Thanks for this important comment. We acknowledge that the rationale for chosing 60 Hz might have not been sufficiently explained in the original version of the manuscript. In the revised introduction, we now clarify and expand the rationale for selecting 60 Hz. We emphasize that while 60 Hz has been tested in auditory and electrical protocols, its use in visual entrainment, especially over multiple weeks, remains largely unaddressed. Our study aims to explore this area. This point has been explicitly stated in the revised introduction (page 5 lines 97-105).

We also thank the Reviewer for pointing us to key references in this field. These are very useful to provide context to our findings and support an interesting role of 60 Hz modulation in humans. Accordingly, we now reference:

Manippa et al. (2024), who reported reduced intrusion errors and enhanced cognitive control in healthy adults following 60 Hz auditory stimulation,

Nomura et al. (2019) who demonstrated that 60 Hz tACS over the dorsolateral prefrontal cortex (DLPFC) enhances declarative long-term memory (LTM).

We still note that much less literature exists on 60 Hz entrainment (especially driven by flickering light) as compared to 40 Hz, which has been extensively studied. Given its functional relevance in executive and memory related processes, its role in PNN remodelling (based on our preclinical data) and the paucity of data currently existing, we think there is a strong rationale for exploring the effects on 60 Hz in humans further.

2. At least one lower frequency should have been tested as a control. In the cited animal study (Venturino et al., 2021), 60 Hz—compared to 40 Hz—had a sp

---

## [Decision Letter · Decision Letter 1]

20 Aug 2025

PONE-D-25-10411R1Exploring Neural Entrainment and Synchrony in Response to Repeated 60 Hz Flickering White Light in Healthy VolunteersPLOS ONE

Dear Dr. ferretti,

Thank you for submitting your manuscript to PLOS ONE. After careful consideration, we feel that it has merit but does not fully meet PLOS ONE’s publication criteria as it currently stands. Therefore, we invite you to submit a revised version of the manuscript that addresses the points raised during the review process.

Please, address the following minor concerns:- In the abstract (line 35), there appears to be a typo: "flicking light across multiple cortical regions" should likely read "flickering light".

- The naming of temporal channels is inconsistent throughout the manuscript and figures (e.g., sometimes Tp7-Tp8, Tp5-Tp6, or T5-T6). Please, unify this notation.

- In the caption for SI Figure 6 (lines 1425-1427), the statement "The observed statistically significant differences are driven primarily by the large sample size (n), rather than any meaningful biological variation..." could be misleading. Please, clarify this point in the text to avoid confusion.

We look forward to receiving your revised manuscript.

Kind regards,

Gennady S. Cymbalyuk, Ph.D.

Academic Editor

PLOS ONE

Journal Requirements:

Reviewers' comments:

Reviewer's Responses to Questions

**Comments to the Author**

1. If the authors have adequately addressed your comments raised in a previous round of review and you feel that this manuscript is now acceptable for publication, you may indicate that here to bypass the “Comments to the Author” section, enter your conflict of interest statement in the “Confidential to Editor” section, and submit your "Accept" recommendation.

Reviewer #1: All comments have been addressed

Reviewer #2: All comments have been addressed

2. Is the manuscript technically sound, and do the data support the conclusions?

Reviewer #1: Yes

Reviewer #2: Yes

3. Has the statistical analysis been performed appropriately and rigorously? 

Reviewer #1: Yes

Reviewer #2: Yes

4. Have the authors made all data underlying the findings in their manuscript fully available?

Reviewer #1: Yes

Reviewer #2: Yes

5. Is the manuscript presented in an intelligible fashion and written in standard English?

Reviewer #1: Yes

Reviewer #2: Yes

6. Review Comments to the Author

Reviewer #1: The authors have addressed my previous comments and improved the manuscript accordingly. A few minor suggestions remain:

- In the abstract (line 35), there appears to be a typo: "flicking light across multiple cortical regions" should likely read "flickering light".

- The naming of temporal channels is inconsistent throughout the manuscript and figures (e.g., sometimes Tp7-Tp8, Tp5-Tp6, or T5-T6). I advise unifying this notation.

- In the caption for SI Figure 6 (lines 1425-1427), the statement "The observed statistically significant differences are driven primarily by the large sample size (n), rather than any meaningful biological variation..." could be misleading. I suggest clarifying this point in the text to avoid confusion.

Reviewer #2: The majority of my concerns have been adressed and additional analyses included. As I understand that addiing further partcipants will not change the result I approve publication.

7. PLOS authors have the option to publish the peer review history of their article (what does this mean? ). If published, this will include your full peer review and any attached files.

**Do you want your identity to be public for this peer review?** For information about this choice, including consent withdrawal, please see our Privacy Policy .

Reviewer #1: No

Reviewer #2: No

---

## [Author Response · Author response to Decision Letter 2]

26 Aug 2025

- In the abstract (line 35), there appears to be a typo: "flicking light across multiple cortical regions" should likely read "flickering light".

We thank the Reviewer for highlighting this typo. This has been corrected.

- The naming of temporal channels is inconsistent throughout the manuscript and figures (e.g., sometimes Tp7-Tp8, Tp5-Tp6, or T5-T6). Please, unify this notation.

Thanks for pointing us to this inconsistency. We have now edited the text throughout the manuscript, using T5-T6 to refer to temporal channels.

- In the caption for SI Figure 6 (lines 1425-1427), the statement "The observed statistically significant differences are driven primarily by the large sample size (n), rather than any meaningful biological variation..." could be misleading. Please, clarify this point in the text to avoid confusion.

We concur with the Reviewer that this sentence might have been misleading. We have now edited as follows: ´The statistically significant differences in PLV observed under constant light conditions are likely attributable to the large number of channel pair comparisons (n) rather than meaningful biological variation, since the average PLV values across groups and days remain comparable.´

---

## [Editor Report · Decision Letter 2]

29 Aug 2025

Exploring Neural Entrainment and Synchrony in Response to Repeated 60 Hz Flickering White Light in Healthy Volunteers

PONE-D-25-10411R2

Dear Dr. ferretti,

We’re pleased to inform you that your manuscript has been judged scientifically suitable for publication and will be formally accepted for publication once it meets all outstanding technical requirements.

Kind regards,

Gennady S. Cymbalyuk, Ph.D.

Academic Editor

PLOS ONE
---

## [Editor Report · Acceptance letter]

PONE-D-25-10411R2

PLOS ONE

Dear Dr. Ferretti,

I'm pleased to inform you that your manuscript has been deemed suitable for publication in PLOS ONE. Congratulations! Your manuscript is now being handed over to our production team.

Kind regards,

on behalf of

Dr. Gennady S. Cymbalyuk

Academic Editor

PLOS ONE